# Exploring the vibrational series of pure trilobite Rydberg molecules

Max Althön[1], Markus Exner[1], Richard Blättner[1] & Herwig Ott [1] ✉

In trilobite Rydberg molecules, an atom in the ground state is bound by electron-atom scattering to a Rydberg electron that is in a superposition of high angular momentum states. This results in a homonuclear molecule with a permanent electric dipole moment in the kilo-debye range. Trilobite molecules have previously been observed only with admixtures of low-$l$ states. Here we report on the observation of two vibrational series of pure trilobite Rubidium-Rydberg molecules that are nearly equidistant. They are produced by three-photon photoassociation and lie energetically more than 15 GHz below the atomic 22F state of rubidium. We show that these states can be used to measure the electron-atom scattering length at low energies in order to benchmark current theoretical calculations. In addition to measuring their kilo-Debye dipole moments, we also show that the molecular lifetime is increased compared to the 22F state due to the high-$l$ character. The observation of an equidistant series of vibrational states opens the way to observe coherent molecular wave packet dynamics.

Creating controllable molecules at ultralow temperatures offers a pathway to engineered ultracold quantum chemical reactions[1–4] and tests of fundamental physics and symmetries[1]. Molecules that possess sizeable electric dipole moments can be controlled by external electric fields making them candidates for quantum information processing[5,6] and the production of strongly correlated many-body systems[7,8]. For dipolar molecules with multiple vibrational states electric field pulses have been proposed to create superposition states[9] and observe coherent wave-packet dynamics.

Ultralong-range Rydberg molecules (ULRMs)[10–12] are a platform for creating such dipolar molecules in ultracold environments. In these molecules, a neutral ground state atom is trapped inside the giant electronic wavefunction of a Rydberg state by a binding mechanism stemming from the electron-ground state scattering interaction. ULRMs have been found to be an ideal testbed for low-energy electron-ground state scattering[13–16] and could be used for the investigation of diabatic coupling schemes in molecules[17]. They can also be used as a starting point for the creation of ultracold anions[18]. Homonuclear ULRMs can have a permanent electric dipole moment due to the distinguishability of the ground state and Rydberg electron[19]. For ULRMs corresponding to low-$l$ ($S$, $P$, $D$) Rydberg states this can reach about

one Debye. There are also two classes of molecules emerging from the mixing of multiple high-$l$ Rydberg states. These so-called butterfly[20] and trilobite molecules can have dipole moments on the order of kilo-Debye[21], which are in special cases even larger than the bond length[20].

Due to the high-$l$ nature of their electronic wave function trilobite molecules are in general not accessible with standard one- or two-photon photoassociation. Nevertheless, states with significant trilobite admixture have been produced via two-photon excitation both in Cs[21] and Rb[22]. In Cs, the almost integer quantum defect of $S$ states leads to a mixing with the high-$l$ states, whereas in Rb a sizable admixture only exists for a specific principal quantum number, where the splitting between the $S$ state and the high-$l$ manifold matches the ground state hyperfine splitting.

Here, we use three-photon excitation to produce pure trilobite molecules in Rb over a wide range of frequencies and characterize their binding energies, lifetimes, and dipole moments. We observe two vibrational series which are energetically split because of different angular momentum couplings and show that their lifetimes exceed that of the adjacent 22F state. Because the excited molecules are of pure trilobite type we find dipole moments which are almost as large as the bond length.

[1]Department of Physics and Research Center OPTIMAS, Rheinland-Pfälzische Technische Universität Kaiserslautern-Landau, 67663 Kaiserslautern, Germany. ✉e-mail: herwig.ott@rptu.de

## Results

ULRMs form due to the elastic scattering interaction of the Rydberg electron with a neutral ground-state atom. To describe the scattering process, Fermi pseudo potentials[23,24] with energy-dependent scattering lengths are used. In atomic units (a.u.), the interaction is given by

$$
\hat{V} = A\hat{\mathbf{s}_2} \cdot \hat{\mathbf{I}} + \sum_{S,T} 2\pi \hat{\mathbb{P}}_{S,T} a_s^{S,T}(k)\delta(\mathbf{R} - \mathbf{r})
$$
$$
+ 6\pi \hat{\mathbb{P}}_{S,T} \left( a_p^{S,T}(k) \right)^3 \delta(\mathbf{R} - \mathbf{r}) \overleftarrow{\nabla} \cdot \overrightarrow{\nabla},
$$

(1)

where $\mathbf{r}$ is the position of the Rydberg electron and $\mathbf{R}$ is the internuclear axis between the Rydberg core and the ground state atom, as shown in Fig. 1a. The s- and p-wave scattering lengths $a_{s/p}^{S,T}$ depend on the spins of the electrons resulting in singlet and triplet channels with the according projection operators $\hat{\mathbb{P}}_{S,T}$. To explain the observed spectra the hyperfine interaction of the ground state atom $A\hat{\mathbf{s}_2} \cdot \hat{\mathbf{I}}$ needs to be taken into account. The scattering interaction depends on the Rydberg electron's momentum $k$ relative to the ground state atom, which is calculated semi-classically for every internuclear distance $R$ as $k = \sqrt{-1/n^2 + 2/R}$ (in a.u.). For the $k$-dependence of the singlet scattering lengths, we use data provided by I. Fabrikant[25]. For the triplet channels, we employ a model potential consisting of a polarization potential with an inner hard wall at a variable distance from the ground state atom which captures the short-range physics[13,26]. By varying the position of the hard wall the scattering interaction can be tuned.

We diagonalize the Hamiltonian given in[27], which includes spin-orbit coupling of the p-wave scattering, at each internuclear distance. We consider a finite basis set consisting of two hydrogenic manifolds below the state of interest ($n = 22$) and one manifold above it, as well as all states that lie energetically in between. This basis set was chosen because previous studies show that it most closely matches an alternative Green's function method of calculating the potential energy curves[13,27]. The resulting Born-Oppenheimer potential energy curves are shown in Fig. 2. The energy curves belong to different types of

molecules and show avoided crossings where the molecular character changes. Of particular interest for this work is the crossing between the trilobite and butterfly curves. For the chosen $n = 22$ this results in three mutually shifted potential wells which are particularly deep and nearly harmonic supporting multiple vibrational states.

While the lower potential well can be assigned to the $F = 1$ ground state and triplet s-wave scattering, the middle potential curve consists of a mixture of the two hyperfine states and shows both singlet and triplet s-wave scattering[14,28,29]. This mixture is due to the interplay between the hyperfine interaction and the electron scattering interaction, as both depend on the spin state of the ground state electron. The upper potential well for the $F = 2$ ground state cannot be excited in our experiment, as we prepare the sample in the $F = 1$ state. The bound states are then calculated from the potential curves with a so-called shooting method[30] by analyzing the density of states when varying the inner boundary condition[15].

To photoassociate the trilobite Rydberg molecules in the lower two wells we use a three-photon setup with lasers at 780 nm, 776 nm, and 1288 nm. This allows us to couple to the 22F state, which makes up about 3% of the electronic state. The first two lasers are blue-detuned to the intermediate states (126 MHz above $5P_{3/2}F = 2$ and 84 MHz above $5D_{5/2}F = 1$). The three-photon Rabi frequency is $2\pi \times 250$ kHz. Our sample consists of $^{87}$Rb atoms in the $F = 1$ ground state prepared in an optical dipole trap at 1064 nm. The peak density is $4 \times 10^{13}$ cm$^{-3}$ and the temperature is about 150 μK. After the sample preparation we perform 800 pulse sequences. At the start of each pulse sequence, the dipole trap is switched off. This is mainly because the 1064 nm light efficiently ionizes any remaining population in the $5D_{5/2}$ intermediate state and thus leads to a large number of background ions. Afterwards, the three excitation lasers are pulsed in for 1 μs. To detect the Rydberg excitations an extraction field is switched on after the excitation pulse and after a variable delay time, a $CO_2$ laser pulse ionizes all Rydberg states. The resulting ions are guided via a reaction microscope[31–33] to a space- and time-resolved multi-channel plate detector. This allows us to measure the momentum of the Rydberg core prior to ionization. Note that the recoil upon ionization with the $CO_2$ laser is negligible. After the ionization pulse, the dipole trap is switched on again to recapture the remaining atoms. Because the dipole trap has to be switched off during excitation this release-recapture scheme enables more efficient molecule excitation than e.g. one longer excitation pulse. It also allows for precise timing between excitation and ionization.

Because of the large dipole moments of the trilobite molecules, precise electric field compensation during the excitation pulses is necessary. To achieve this we use the momentum imaging capabilities of our reaction microscope. In this field compensation measurement, the atoms are ionized and accelerated in the residual electric field for a variable wait time. Afterwards, we measure the momenta of the ions and extract the electric field from the linear dependence on the wait time.

Figure 3 shows the molecular spectrum red detuned to the $22F_{7/2}$ state covering the two trilobite potential wells. We observe a vibrational series of up to $v = 6$ in each of the potential wells, which are equally spaced. The anharmonicity is less than 10 percent, confirming the harmonic oscillator shape of the potential wells. Thereby, the highest observed vibrational states are consistent with the theoretically predicted potential depths. Next, we analyze in detail the position of the vibrational states and the conclusions one can draw for the molecular potential. Inspecting the different terms in Eq. (1) shows that the molecular potential for the trilobite curve is proportional to the respective scattering length. High-precision molecular Rydberg spectroscopy is therefore a tool to determine the electron-atom scattering lengths. Because of the crossing with the butterfly curves, both the triplet p-wave as well as the dominant triplet s-wave scattering channels have to be considered. However, because of the relative

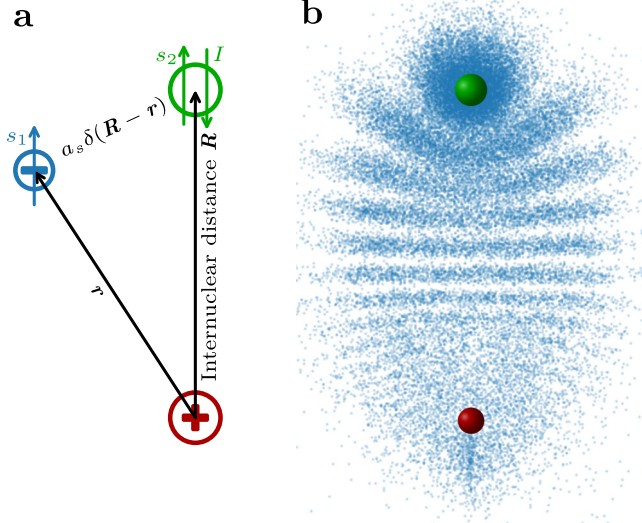

**Fig. 1 | Sketch of a trilobite Rydberg molecule. a** Sketch of a Rydberg molecule. The coordinates of the Rydberg electron (blue) and ground state atom (green) relative to the Rydberg core (red) are denoted with black arrows. The relevant spins are that of the Rydberg electron $s_1$, the electron of the ground state atom $s_2$ and the nuclear spin of the ground state atom $I$. **b** Sketch of a trilobite molecule. The Rydberg core and the ground state atom are shown (with exaggerated size) as red and green spheres respectively. The electronic probability density projected to 2D is indicated by the density of blue dots.

 

insensitivity to the triplet p-wave scattering, we cannot accurately determine the corresponding scattering length and instead use previously measured values[13]. Since the mixed trilobite has a small singlet admixture, the splitting of the two potential curves also depends on

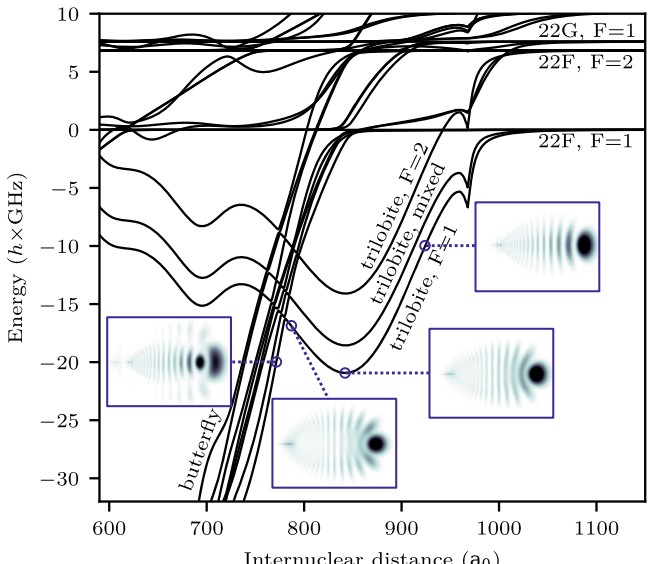

**Fig. 2 | Born-Oppenheimer potential energy curves resulting from the electron-ground state scattering.** The energy zero is the asymptotic pair state energy of a $22F_{7/2}$ state and an F = 1 ground state atom. The three labeled trilobite potentials result from different couplings of the spins $s_1$, $s_2$, and $I$. While the curves labeled F = 1 and F = 2 exhibit triplet scattering, the mixed curve has a mixture of singlet and triplet scattering, which in turn means a superposition of F = 1 and F = 2 in the ground state atom. The kink at about $970\,a_0$ is due to the semi-classical calculation of the electron momentum $k$. Insets: Electronic probability density for the trilobite molecule shown at three different internuclear distances. At about $780\,a_0$ the trilobite curves are crossed by the butterfly potentials, which result from a shape resonance in the p-wave scattering. The leftmost inset shows the electronic structure of a butterfly molecule.

the singlet s-wave scattering length. Singlet scattering lengths calculated by a two-active-electron model[25] fit the observed splitting well over the entire range of relevant internuclear distances, with a value of $a_s^S(k = 0.0175\,\text{a.u.}) = (6.3 \pm 0.5)\,a_0$ at the position of the potential minimum. To determine the triplet s-wave scattering length, we employ an iterative procedure. The position of the inner hard wall $r_0$ of the model potential is varied and the Schrödinger equation is solved for the altered potential to determine the scattering length. With this changed scattering length, the molecular Hamiltonian is diagonalized and the vibrational bound states in the new potential energy curve are calculated. This is repeated until the deviation to the measurement is minimized. The only free parameter is hereby $r_0$. We find a value of $a_s^T(k = 0.0175\,\text{a.u.}) = (-7.8 \pm 0.3)\,a_0$ at the position of the potential minimum. The triplet s-wave scattering length at zero momentum can then be calculated from $r_0$ and the polarizability of the perturber $\alpha = 320.1\,\text{a.u.}[34]$ with $a_s^T(k = 0) = \sqrt{\alpha}\cot(\sqrt{\alpha}/r_0)[35]$. This results in $a_s^T(k = 0) = (-14.6 \pm 0.3)\,a_0$.

A peculiar property of trilobite Rydberg molecules is their large permanent electric dipole moment. This stems from the large concentration of the electron density at the position of the ground state atom (see Fig. 1b). The dipole moments are measured by applying an electric field and observing the broadening of the molecular line. Close to zero field, the full width at half maximum obtained from a Lorentzian fit to the molecular peaks is 7 MHz. As the rotational splitting (60 kHz) can therefore not be resolved in our experiment, we fit the spectra with the convolution of a Lorentzian with a step function of width $2dE[21]$. From the fitted widths $dE$ for different electric fields we can deduce the dipole moment as shown in Fig. 4. We find electric dipole moments up to ~1700 Debye, which corresponds to 0.8 times the internuclear distance. This reflects the highly efficient binding mechanism, which accumulates the electron density at the location of the ground state atom. For the theoretical calculation of the dipole moments we write the electronic wavefunction at internuclear distance $R$ on the basis of the unperturbed states

$$\left|\Psi_{\text{mol}}^{(R)}\right\rangle = \sum_i c_i^{(R)}|i\rangle \qquad (2)$$

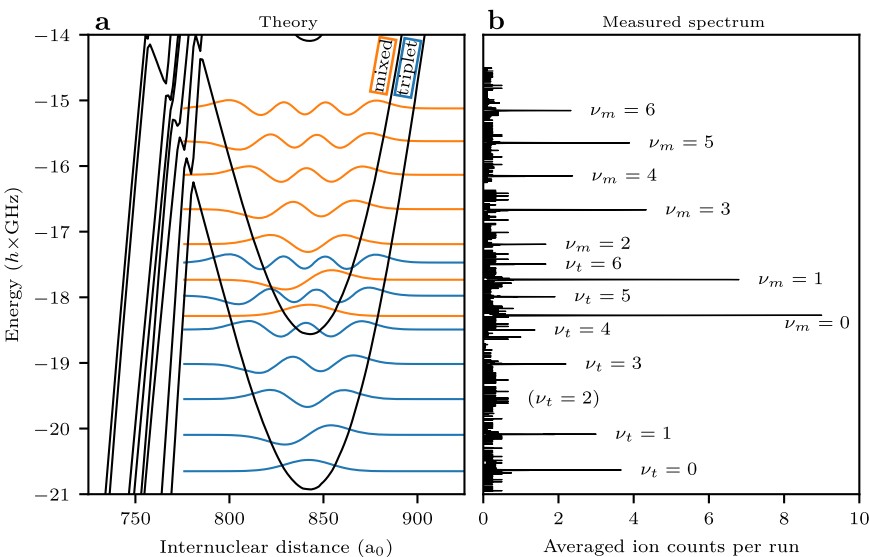

**Fig. 3 | Theoretical potential energy curves and measured spectrum.**
**a** Theoretical potential energy curves for a triplet s-wave asymptote of $a_s^T(k = 0) = -14.6\,a_0$ and the resulting vibrational wavefunctions. The wavefunctions are drawn at their respective binding energies, in orange for the mixed potential and in blue for the triplet F = 1 potential. **b** Measured spectrum. The energy axis is shared with the theoretical potential shown on the left. The peaks are labeled with the vibrational quantum numbers $v_m$ for the mixed potential and $v_t$ for the triplet F = 1 potential.

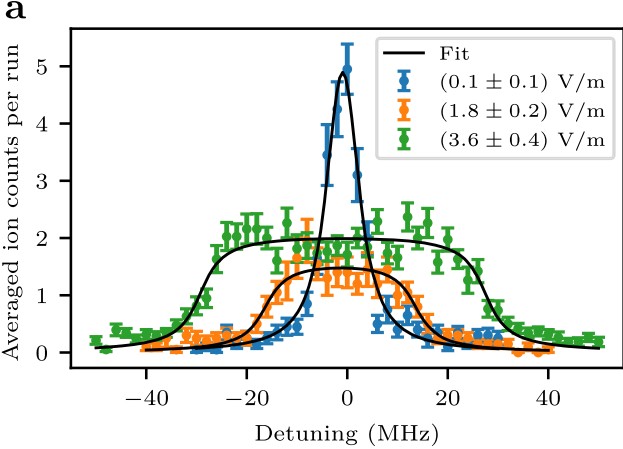

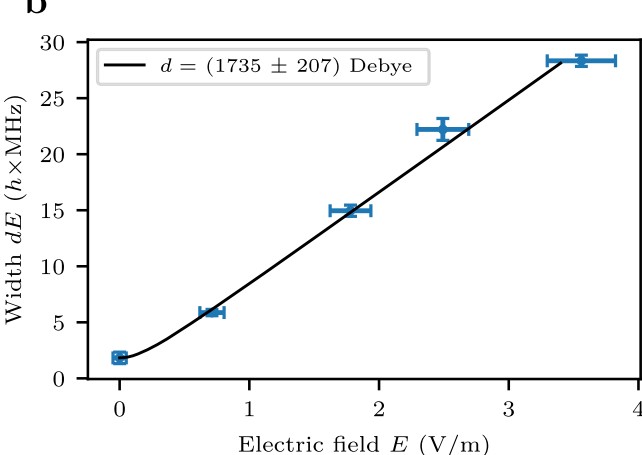

**Fig. 4 | Measurement of the dipole moment for one of the trilobite molecules.**
**a** Spectroscopy of the $v_m = 0$ molecule at -18.274 GHz for different electric fields. The differences in peak heights are due to fluctuations in the sample preparation. The error bars give the standard error of the mean for multiple measurement runs. The data is fitted by the convolution of a Lorentzian with a step function of width $2dE$ (see text). **b** The resulting values of $dE$ from the fit of the spectrum are plotted against the electric field with the error bar of $dE$ calculated from the covariance of the fit. The field error bar is calculated from a field compensation measurement using the reaction microscope. The data is fitted to extract the dipole moment by the function $d\sqrt{E^2 + E_0^2}$ allowing for an offset field $E_0$, which is on the order of 0.1 V/m.

and then integrate over the vibrational wavefunction $\Phi$

$$\langle d \rangle = \int |\Phi(R)|^2 \sum_{i,j} c_i^{(R)*} c_j^{(R)} \langle i|\hat{d}|j\rangle \, dR \tag{3}$$

The experimental and theoretical results for selected vibrational states are presented in Table 1. Experiment and theory are in good agreement, however, the experimentally determined dipole moments are systematically 10 – 15 % larger than the theoretical values. This could be due to an unidentified systematic measurement error or incorrect theoretical dipole matrix elements as small systematic deviations add up due to the many states that contribute to the trilobite wave function. Another possibility is that the semi-classical calculation of the electron momentum $k$ leads to an incorrect prediction of the bond length and the trilobite molecules actually form at larger internuclear distances. One should also consider that any inaccuracy in the description of the interaction (Eq. (1)) results in different coefficients $c_i^{(R)}$ in the state composition (Eq. (2)) and therefore changes the dipole moment.

**Table 1 | Binding energies, dipole moments, and lifetimes of the trilobite rubidium Rydberg molecules**

| | Binding energy ($h \times$ MHz) | | Dipole moment (Debye) | | Lifetime (µs) |
|---|---|---|---|---|---|
| | exp. | theo. | exp. | theo. | exp |
| **22F$_{7/2}$** | **0** | | | | **5.7 ± 0.2** |
| $v_m = 6$ | −15154 ± 2 | −15123 | | 1472 | 7.5 ± 0.5 |
| $v_m = 5$ | −15648 ± 2 | −15622 | 1632 ± 193 | 1477 | 11.0 ± 0.9 |
| $v_m = 4$ | −16152 ± 2 | −16134 | | 1484 | |
| $v_m = 3$ | −16668 ± 2 | −16657 | | 1489 | 13.1 ± 1.0 |
| $v_m = 2$ | −17192 ± 2 | −17190 | | 1495 | |
| $v_m = 1$ | −17730 ± 2 | −17733 | | 1500 | |
| $v_m = 0$ | −18274 ± 2 | −18285 | 1735 ± 207 | 1507 | 14.6 ± 1.0 |
| $v_t = 6$ | −17497 ± 2 | −17472 | | 1472 | |
| $v_t = 5$ | −17990 ± 2 | −17977 | | 1477 | 8.2 ± 0.9 |
| $v_t = 4$ | −18498 ± 2 | −18492 | | 1482 | |
| $v_t = 3$ | −19018 ± 2 | −19017 | | 1489 | 10.6 ± 1.0 |
| $v_t = 2$ | | −19552 | | 1495 | |
| $v_t = 1$ | −20086 ± 2 | −20100 | 1667 ± 198 | 1500 | |
| $v_t = 0$ | −20632 ± 2 | −20650 | 1703 ± 203 | 1505 | 12.0 ± 0.9 |

The binding energy is given with respect to the energy of the 22F$_{7/2}$ state. The theoretical values are calculated with a triplet s-wave asymptote of $a_s^T(k=0) = -14.6\,a_0$. The theoretical radiative lifetime of the 22F$_{7/2}$ state is 6.3 µs. A theoretical prediction of the lifetimes of the molecular states is beyond the scope of this paper.

The second important characteristic of Rydberg molecules is their lifetime. They reflect the different available decay channels, such as spontaneous emission, black-body induced transitions, and molecular decay via tunneling towards shorter internuclear distances, leading to $l$-changing collisions or to associative ionization. The lifetimes are measured by varying the delay time between the excitation and ionization. We then count the number of ions that have zero momentum. This way, we also account for $l$-changing collisions, which result in an ion but come along with a large momentum[36]. Associative ionization results in Rb$_2^+$ which can be distinguished by its larger time-of-flight due to its larger mass. Both associative ionization and $l$-changing collisions stem from a tunneling process out of the potential well into the butterfly potential curve.

The measured lifetimes are given in Table 1. We first note that even the shortest measured lifetime is larger than the lifetime of the atomic 22F Rydberg state and the ground state in each potential well has double its lifetime. This reflects the multitude of involved high-$l$ states in the trilobite molecules, resulting in slower radiative decay. As expected, the tunneling processes are more prominent for higher vibrational states, resulting in a shorter lifetime than the deeply bound vibrational ground states. This is corroborated by a complementary analysis of the data that is not counted towards the lifetime measurement. Looking at either ions with larger momenta stemming from $l$-changing collisions, or Rb$_2^+$ molecular ions an increase in rates of roughly 60% is observed when comparing $v_m = 6$ with the ground state. This proves that tunneling into the butterfly curve is a major decay channel for the higher vibrational states. However, the quantitative dependence on the vibrational quantum number needs further investigation, as e.g. vibronic coupling effects[17] lead to a superposition of butterfly and trilobite states and thus influence the decay dynamics.

## Discussion
Through comparison with theory, singlet and triplet s-wave scattering lengths can be extracted from the measured spectrum. The error of the triplet scattering length in this work is given mainly by the uncertainty associated with the basis choice for the diagonalization[27]. While the basis we use with effective principal quantum numbers

$20 \leq n_{\text{eff}} \leq 23$ was shown to be the closest match to the alternative Green's function approach[27], the choice of basis still impacts the results significantly. We calculate the uncertainty by comparison to the basis that shows the largest discrepancy within the limits $19 \leq n_{\text{eff}} \leq 24$. This results in a relative error of 2%. The experimental results presented here are in principle much more precise, due to a frequency uncertainty of less than 2 MHz relative to the binding energies on the order of 10 GHz. Future theoretical efforts could therefore extract the scattering length from our experimental data to better precision. On the other hand, for the singlet s-wave scattering length, the uncertainty due to basis choice is negligible. Therefore, we can estimate the uncertainty by looking at the sensitivity to changes in the scattering length. For this, we analyze the root-mean-square of the differences between the first five bound states of the mixed potential and their theoretical counterparts. To change this overall deviation by an amount equal to the experimental accuracy of 2 MHz, the vibrational states need to be shifted by about 7 MHz. This corresponds to a change in the singlet scattering length of 0.5 $a_0$ over the entire range of relevant $k$, which we take as the uncertainty. For comparison, to shift the vibrational states of the triplet potential by 7 MHz a change in the triplet s-wave scattering length of only 0.003 $a_0$ is sufficient, illustrating the difference in sensitivity due to the much smaller singlet admixture.

Our result for the triplet s-wave scattering length asymptote is on the lower end of previous experimental values ($-14.7\,a_0$ to $-16.1\,a_0$)[13–15,37] measured at $k$ values near zero. It is also evident that these experimental results vary by up to ten percent. In part, this is due to the aforementioned uncertainties of the diagonalization, but they cannot fully explain the observed discrepancy. One possibility is that the $k$-dependence of the scattering length predicted by the theoretical models is incorrect. This would lead to different zero-energy extrapolations of the scattering length because the molecules used in the various experiments represent different ranges of $k$. We note here that previous ab initio calculations of the scattering lengths[25,38] cannot explain the measured spectrum and therefore do not present an alternative. To resolve this, measurements of trilobite spectra at different principal quantum numbers can be used to probe different ranges of the electron momentum and thus present an opportunity to map out this dependency. This however necessitates a more sophisticated theoretical approach[39]. With such measurements, one can also test whether the semiclassical calculation of $k$ plays a role in the discrepancy. In fact, if the actual electron momentum is assumed to be about 10 % larger than the semiclassical calculation, the binding energy of the triplet trilobite can be brought into line with the previously measured scattering length asymptote of Engel et al.[13], who use the same theoretical model to extract the scattering length. Therefore, these exotic molecules could lead to a better theoretical understanding of the more general process of electron-atom scattering.

In conclusion, we have measured two vibrational series of pure trilobite Rydberg molecules by employing three-photon photoassociation. With this method, the creation of trilobite molecules in any element that has a negative s-wave scattering length should be possible, as the quantum defects for the admixed atomic state (here F-state) are rather small and the coupling with the trilobite state is sizable. We find kilo-Debye dipole moments and lifetimes longer than the coupled atomic state. The observed spectra can be theoretically explained by adjusting the triplet s-wave scattering length. While the resulting agreement is excellent, the extrapolated scattering length asymptote disagrees somewhat with previous measurements and merits further theoretical and experimental work.

Additionally, for higher principal quantum numbers vibronic coupling effects between the trilobite and butterfly curves become more pronounced and these molecules could serve as a benchmark for theoretical calculations[17]. It has also been predicted that at certain principal quantum numbers, conical intersections essentially stop the

$l$-changing collision processes[40], which could be checked with our reaction microscope. Finally, the shape of the potential well is suitable to study coherent wave packet dynamics. Note that the quality factor of the potential well is $Q = \frac{|E(\nu_t = 1) - E(\nu_t = 0)| \cdot \tau}{h} \approx 3 \times 10^4$ with a lifetime of $\tau = 10\,\mu$s. Using ns and ps laser pulses in a pump-probe scheme provides the required time resolution for such experiments.

## Methods
### Three-photon excitation
To photoassociate the trilobite states we use a three-photon setup which can couple the 5S ground state of Rubidium to $n$P or $n$F states. The first excitation laser (780 nm) is a distributed-feedback diode laser, the other two (776 nm and 1288 nm) are external cavity diode lasers. All three lasers are stabilized and controlled in the same way. We split the light into two pathways, one going to the experiment and one being used for stabilization. In the stabilization path, an electro-optic modulator is used to create sidebands on the light. These sidebands are then stabilized with a Pound-Drever-Hall lock[41] to an ultralow-expansion cavity with a free spectral range of 1497.8 MHz which serves as an extremely stable frequency reference. By varying the frequency of the electro-optic modulator the laser frequency can be tuned relative to the cavity modes with high precision and accuracy. The 780 nm and 776 nm lasers remain at a fixed frequency, while the 1288 nm laser is tuned. The $22\text{F}_{7/2}$ resonance is determined experimentally and serves as the energy zero. To reach the large detunings necessary to associate the trilobite molecules, multiple free spectral ranges of the cavity need to be crossed. For accurate frequency determination, it is therefore paramount to know the free spectral range precisely. Since we know the free spectral range to a precision of 100 kHz, the maximum frequency uncertainty encountered is $\frac{21\,\text{GHz}}{1.4978\,\text{GHz}} \cdot 100\,\text{kHz} \approx 1.4\,$MHz, which is smaller than the 2 MHz frequency steps we chose to record the spectrum.

### Vibrational bound states
Once the potential energy curves are found from the diagonalization, the vibrational states are obtained numerically with a so-called shooting method. First, an inner and an outer boundary to the potential energy curve are set. The outer boundary is chosen such that it is firmly in the classically forbidden region for the vibrational states of interest. Next, wavefunctions corresponding to an initial energy grid are calculated by numerical integration of the Schrödinger equation with Numerov's method[42]. It is checked at which energies the wavefunctions are close to physically valid (zero at the outer boundary), meaning that there is a sign change between two energetically adjacent wavefunctions at the outer boundary. This information is used to refine the energy grid and the process is repeated until a precision target is reached. This method is sensitive to the position of the inner boundary. To get accurate results the inner boundary position is varied over a large range of internuclear distances and the density of found states is analyzed. Peaks in the density of states correspond to bound states in the potential.

### Determination of scattering lengths
The model potential used to calculate the $k$-dependent scattering lengths is given by

$$V = \begin{cases} \frac{-\alpha_p}{2r^4} + \frac{L(L+1)}{2\mu r^2} + \delta_{L1}\delta_{S1}\partial_r \left(\frac{-\alpha_p}{2r^4}\right)\frac{J(J+1)-4}{4\gamma^2 x}, & r \geq r_0 \\ \infty, & r < r_0 \end{cases} \quad (4)$$

with the polarizability of the perturber $\alpha_p = 320.1$ a.u.[34] and the inverse of the finestructure constant $\gamma$. It consists for the s-wave ($L = 0$) of a polarization potential and an inner hard wall at $r_0$. For the p-wave ($L = 1$) there is an additional centrifugal potential and a potential due to the finestructure in the triplet scattering ($S = 1$). By numerically solving the Schrödinger equation for a specific $k$ in this model potential one can

determine the scattering phaseshift $\delta(k)$ by comparing the solution to the wavefunction of a free particle with momentum $k$. The s-wave scattering length is calculated as $a_s = \tan(\delta_s(k))/k$, while the p-wave scattering volume is given by $(a_p)^3 = \tan(\delta_p(k))/k^3$. To fit the scattering length to the experimental data, the position of the inner hard wall $r_0$ is varied, which is the only free parameter. With this changed scattering length, the molecular Hamiltonian is diagonalized and the vibrational bound states in the new potential energy curve are calculated. This is repeated until the deviation to the measurement is minimized.

## Data availability

The data generated in this study is available under https://doi.org/10.26204/data/6.

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

## Acknowledgements
We would like to thank Frederic Hummel, Peter Schmelcher and Matt Eiles for helpful discussions. This project is funded by the Deutsche Forschungsgemeinschaft (DFG, German Research Foundation) - project number 316211972 and 460443971.

## Author contributions

M.A., M.E., R.B. performed the experiments. M.A. and M.E. analyzed the data. M.A. performed the theoretical calculations and prepared the initial version of the manuscript. H.O. conceived and supervised the project. All authors contributed to the data interpretation and manuscript preparation.

## Funding

## Competing interests
The authors declare no competing interests.
