## [Peer Review File · Nature Communications]

Exploring the vibrational series of pure trilobite Rydberg moleculesReviewers' Comments:

Reviewer #1:

Remarks to the Author:

Referee comment:

In the manuscript, Althön et al. report the observation of two vibrational series in the trilobite Rydberg molecules formed by three-photon photoassociation. The experimental observation is compared to the theoretical prediction, resulting in a good match. The manuscript is overall well-written and the results are scientifically sound. However, the manuscript can use some updates to enhance its strength and help readers understand the results better. I can recommend its publication after my few concerns are addressed.

General comment

It was not immediately clear to me how the main experimental result (Fig. 3b) and other results have been obtained. I think more details on the experimental setup would be useful in general for this manuscript. (See below) .

Similarly, the theoretical calculation/analysis described around Line 151- Line 197 would be better if more details are given. Namely, what are the exact procedures to obtain these theoretical values? Is it just a curve fit or does it involve some sort of iteration of adjusting the scattering length value, diagonalizing Hamiltonian, and solving vibrational states?

Below is a list of my more specific comments.

(1) Could you add discussion on why "the observed states kilo-Debye dipole moments despite its relatively low principal quantum number"? (related to line 59)

(2) Line 88: Do you have justification for including only these three manifolds? With these three manifolds, how many total states (including spins and AMs) do you have in the basis set?

(3) Could you discuss why $n=22$ is particularly interesting? (Line 90)

(4) The author briefly describes the three-photon PA. Presumably, the authors know the frequency of these three lasers with better precision. Could you add these numbers? How are the energy levels of the Rydberg states determined? (Fig. 2b y-axis). I assume they are determined by the sum of the photon energies compared to the known $22F$ state. I think it is useful to know the accuracy of the used laser frequencies as well if you have them, because the total uncertainty of the three lasers combined becomes the uncertainty of the y-axis. Are these three lasers all CW lasers? Is 1288nm also ECDL?

(5) The 800 excitation pulses (Line 121): I am a little confused as to whether these are PA pulses or ionization pulses. It seems to make more sense to me if they are ionization pulses. Could you clarify this? (I think I was confused because of the order of those sentences.) If PA is actually pulsed, please explain why that is advantageous.

(6) Could you quantify the equidistant nature of the vibrational series? Most of the peaks are clearly identified, so I think it is possible to get the frequency difference between these peaks. The lower triplet potential (theory) seems to have slight asymmetry. Does it show up in the level spacing and does it match the expectations?

(7) Do you know why $v_t=2$ peak is weak?

(8) To me, by just looking at the potential curve, $v_t=7$ state might exist. Potentially there is a corresponding peak in the experimental data as well. Is this possible?

(9) Given my previous comment, I don't know if the statement that the highest vibrational state coincides with the crossing... (Line 146) is a strong argument. I agree that the observation is not contradictory to the potential curve.

(10) Line 161: The splitting seems to be R-dependent. Does this singlet s-wave scattering length reproduce the splitting over the entire range of R? What is the value of the scattering length?

(11) Could you describe how you found the J=0 shape resonance?

(12) Line 171: As I commented at the beginning, I am a little confused after all as to what exactly the procedure is for determining the scattering length. If I understand correctly, the scattering length is k-dependent and thus there seems to be too many free parameters to determine for the experimentally obtained information. Do you assume any functional form for the scattering length as a function of k?

(13) What is the exact procedure of extrapolating the scattering length to $k=0$?

(14) I think the majority of the paragraph (Line 182 - Line 197) is actually discussion for future studies rather than the result. I suggest you move this part (or maybe the entire paragraph line 151- line 197) to Discussion section and merge it to the partially overlapping sentences (Line 268 - Line 281)

(15) Line 204: Could you provide numbers for the splitting and the resolution?

(16) Line 215: How do you calculate the coefficient $C_i(R)$? Is this the same procedure as what you used to calculate the PEC?

(17) Line 236- Line 238: Could you explain what you meant by this sentence? I assume TOF (undefined) stands for time-of-flight.

(18) If the lifetime is indeed affected by the tunneling, do you see an increase in the signal for ions with larger momentum? (Maybe you didn't collect this signal, but I am curious.) I imagine the rate of this tunneling (rate) is sensitive to v near the top of the potential well. Is it possible to use this to check the well depth?

(19) Table 1: The experimentally measured dipole moment seems to be consistently bigger than the theoretical values. Do you have any explanation for this?

(20) Line 291 How did you estimate the quality factor? Which numbers did you use for this estimate?

Reviewer #2:

Remarks to the Author:

This paper describes a theoretical/experimental study of the vibrational structure of pure rubidium ultralong-range trilobite dimer Rydberg molecules. The paper is well organized and well written. The experimental approach is sound and the results appear to be of good quality and are supported by theory. The predicted series of near-equally-spaced vibrational levels is observed, consistent with a near-harmonic potential well. The results are then used to evaluate the triplet s-wave scattering length and the energy of the p-wave shape resonance. Their lifetimes and dipole moments are also determined. This is a nice piece of work that definitely merits publication, although I do have a few minor comments:

* The authors comment that bond states are calculated with a "shooting method." This is not a term I am familiar with. Perhaps they could use a couple of sentences to expand upon it.

*The authors comment that they find "electric dipole moments of up to 1735 Debye." Given the sizable errors listed in Fig. 4, surely it would be better to state "up to ~ 1700 Debye."

* The authors discuss resolving the discrepancy between previously-reported scattering lengths and their measurements by using different values of n . How easy is this to do? Would their existing lasers allow this? Such measurements would certainly strengthen the paper and further

justify the use of long-range dimers for detailed study of electron-atom scattering.

In summary, this work is of high quality and is certainly worthy of publication. My only concern is that it might not be sufficiently novel or of broad general interest to appeal to readers of Nature Communications. Ultralong-range Rydberg molecules have been the subject of many earlier studies and their large electric dipole moments and their use to extract scattering lengths discussed. Also the observation of long lifetimes is hardly surprising given their higher l character.

First of all, we would like to thank the reviewers for their insightful comments and questions. Our answers are included below.

Regarding the revised paper, changes or additions are highlighted in blue. Sections of the text that were removed are crossed out. We have also included a version of the paper, where the changes are not highlighted.

Reviewer 1:

(0) Similarly, the theoretical calculation/analysis described around Line 151- Line 197 would be better if more details are given. Namely, what are the exact procedures to obtain these theoretical values? Is it just a curve fit or does it involve some sort of iteration of adjusting the scattering length value, diagonalizing Hamiltonian, and solving vibrational states?

- We have added some more information to the main text. Also, in the methods section, we have added a more detailed description of the procedure. The procedure is indeed iterative. The scattering length is adjusted by slightly adjusting the position inner hard wall in the model potential. For each k , the Schrödinger equation for this adjusted model potential is then solved and the resulting wavefunction is compared to the wavefunction of a free particle to obtain the phaseshift (and therefore the scattering length). With this adjusted scattering length $a(k)$ the Hamiltonian is diagonalized and the vibrational states are found and compared to the measurement. The procedure is repeated until the root-mean-square deviation between measurement and theory is minimized. We emphasize that the fit procedure only has one free parameter (the position of the inner hard wall).

Below is a list of my more specific comments.

(1) Could you add discussion on why “the observed states kilo-Debye dipole moments despite its relatively low principal quantum number”? (related to line 59)

- The high dipole moment at a low principal quantum number is because we excite pure trilobite molecules compared to previous measurements where states with trilobite admixture were excited. We have changed the text to make this more clear.

(2) Line 88: Do you have justification for including only these three manifolds? With these three manifolds, how many total states (including spins and AMs) do you have in the basis set?

- In total, four manifolds are included ($n=20, 21, 22, 23$) as well as all of the states with $l \leq 4$ in between. This corresponds to 1152 basis states. We chose the basis because a theoretical study (Eiles et al. 2018 [1]) shows that it most closely matches the alternative Green’s function method of calculating the potential energy curves. This finding is corroborated by Engel et al. 2019 [2] who also compared the Green’s function method with diagonalization of different basis sizes and came to an analogous conclusion. However, further investigation into this shows that the uncertainties associated with the choice of basis are rather large. We have now included these uncertainties into the values for the scattering lengths and added a paragraph to the discussion section.

(3) Could you discuss why $n=22$ is particularly interesting? (Line 90)

- For $n > 25$ there are two trilobite wells before the butterfly curves cross the trilobite curves. This adds complexity to the spectra. On the other hand, for small n the butterfly crosses the outer well earlier, resulting in shallower and also more asymmetric wells. Therefore, $n=22$ provides a good opportunity to see many vibrational states (due to the well depth) which are nearly equidistant. We have added this to the paper. The other principal quantum will be investigated in the future to gain further insight, e.g. into the k -dependence of the scattering lengths.

(4) The author briefly describes the three-photon PA. Presumably, the authors know the frequency of these three lasers with better precision. Could you add these numbers? How are the energy levels of the Rydberg states determined? (Fig. 2b y-axis). I assume they are determined by the sum of the photon energies compared to the known $22F$ state. I think it is useful to know the accuracy of the used laser frequencies as well if you have them, because the total uncertainty of the three lasers combined becomes the uncertainty of the y -axis. Are these three lasers all CW lasers? Is 1288nm also ECDL?

- We have added the detunings to the intermediate states to the main text, as well as a more detailed description (similar to the following text) to the Methods section. The first excitation laser (780 nm) is a distributed-feedback diode laser, the other two (776 nm and 1288 nm) are ECDL. All three lasers are stabilized and controlled in the same way. We split the light into two pathways, one going to the experiment and one is used for stabilization. In the stabilization path an electro-optic modulator is used to create sidebands on the light. These sidebands are then stabilized with a Pound-Drever-Hall lock to an ultralow-expansion cavity which serves as a extremely stable frequency reference. By varying the frequency of the electro-optic modulator the laser frequency can be tuned relative to the cavity modes with high precision and accuracy. The 780 nm and 776 nm lasers remain at a fixed frequency, while the 1288 nm laser is tuned. The $22F$ ($J=7/2$) resonance is determined experimentally and serves as the energy zero. For simplicity, let's suppose one of the cavity resonances coincides perfectly with the $22F$ ($J=7/2$) state. To determine a frequency on the y -axis in Fig.3b we can then count the number of cavity resonances, e.g. 10, multiple this number with the free spectral range of the cavity and then add/subtract the EOM frequency, e.g. +500 MHz to arrive at $10 \cdot 1497.8 \text{ MHz} + 500 \text{ MHz} = 15478 \text{ MHz}$. Since we know the free spectral range of the cavity to about 100 kHz, the maximum error of the y -axis is $21 \text{ GHz} / 1.4978 \text{ GHz} \cdot 100 \text{ kHz} \approx 1.4 \text{ MHz}$. Additionally, the laser frequency of the 1288 nm is double checked with a wavemeter.

(5) The 800 excitation pulses (Line 121): I am a little confused as to whether these are PA pulses or ionization pulses. It seems to make more sense to me if they are ionization pulses. Could you clarify this? (I think I was confused because of the order of those sentences.) If PA is actually pulsed, please explain why that is advantageous.

- During excitation, the dipole trap has to be switched off for two reasons. The first and main reason is that the 1064nm light of the trap ionizes the $5D$ state

efficiently and therefore leads to a large number of background ions. The second reason is to avoid AC-Stark shifts. After switching off the dipole trap there is an excitation pulse followed by an ionization pulse. Afterwards, the dipole trap is switched back on to recapture the atom cloud. We repeat this procedure 800 times, so there are 800 excitation pulses and 800 ionization pulses. Because the dipole trap has to be switched off, a release-recapture scheme as implemented here is more efficient than one long excitation pulse. This pulsed scheme also has the advantage that the Rb_2^+ molecular ions can be distinguished (see answer to question (17)) by their longer time-of-flight. We have altered the description to make the experimental sequence clearer.

(6) Could you quantify the equidistant nature of the vibrational series? Most of the peaks are clearly identified, so I think it is possible to get the frequency difference between these peaks. The lower triplet potential (theory) seems to have slight asymmetry. Does it show up in the level spacing and does it match the expectations?

- We have added the measured and theoretical binding energies to the table. There is a slight asymmetry in both potential wells. Both measurement and theory show slight anharmonic behavior, which is approximately the same for both wells. The anharmonicity observed in the experiment is also very similar to the theoretical prediction. For the triplet trilobite the frequency difference between the lowest two vibrational states is 546 MHz (theory: 550 MHz) and the difference between the two highest observed states is 493 MHz (theory: 505 MHz). For the mixed trilobite the frequency difference between the lowest two vibrational states is 544 MHz (theory: 552 MHz) and the difference between the two highest observed states is 494 MHz (theory: 499 MHz).

(7) Do you know why $\nu_t=2$ peak is weak?

- The origin of this is not totally clear to us. We have repeated the measurement with the same result. In general, the different peak heights are determined by the Franck-Condon overlap between two ground state atoms and the molecular wavefunction, as the electronic state only varies slowly with R. Because we work with a thermal sample at non-negligible temperature, there are many possible relative momenta between two ground state atoms which would have to be averaged over appropriately. In addition, the thermal de-Broglie wavelength is on the order of a few hundred bohr radii and therefore not much larger than the vibrational wavefunctions. Thus, it is not trivial to predict the Franck-Condon factors.

(8) To me, by just looking at the potential curve, $\nu_t=7$ state might exist. Potentially there is a corresponding peak in the experimental data as well. Is this possible?

- The theoretical potential would indeed support a $\nu_t = 7$ state. When looking at the measured spectrum in this area in much more detail than the figure allows for, one finds no such peak, considering the standard deviation and the background. We have included a plot in the vicinity of the expected energy (-17 GHz). The frequency difference between each measured point is 2 MHz and the errors are given by the standard error of the mean. The plot also shows the relatively weak $\nu_m = 2$ peak for reference. We have also repeated the measurement to double

check that $\nu_t = 7$ is not present. This means that if the state exists, it most likely has a very short lifetime and is thus broadened to an extent that makes it impossible for us to detect.

(9) Given my previous comment, I don't know if the statement that the highest vibrational state coincides with the crossing... (Line 146) is a strong argument. I agree that the observation is not contradictory to the potential curve.

- We agree that "coincide" might not be the best description in this case and have changed the statement to: "Thereby, the highest observed vibrational states confirm that the well depth for both series is appropriately captured by theory."

(10) Line 161: The splitting seems to be R-dependent. Does this singlet s-wave scattering length reproduce the splitting over the entire range of R? What is the value of the scattering length?

- To good approximation, the triplet and mixed potentials are just shifted by a constant energy, as evident by their almost identical vibrational spacings (see answer to (6)). The singlet s-wave scattering length is k-dependent (and therefore R-dependent) and reproduces the splitting over all relevant R, as the theoretical predictions for the triplet and the mixed potentials fit equally well. We have altered the text to make this more clear. We have also added the value of the singlet scattering length at the potential minimum.

(12) Line 171: As I commented at the beginning, I am a little confused after all as to what exactly the procedure is for determining the scattering length. If I understand

correctly, the scattering length is k -dependent and thus there seems to be too many free parameters to determine for the experimentally obtained information. Do you assume any functional form for the scattering length as a function of k ?

- The scattering length for each k is calculated by numerically solving the Schrödinger equation for the model potential and analyzing the phaseshift between the resulting wavefunction and the wavefunction for a free particle. The functional form of the scattering length is therefore determined by this theoretical model. Comparisons to ab initio calculations and modified effective range theory show that this model produces the correct functional dependence. The advantage of this model potential is that by varying only the position of the inner hard wall, different values for the scattering lengths result. There is therefore only one free parameter.

(11) Could you describe how you found the $J=0$ shape resonance?

- We did this using the same iterative procedure as described in the answers to (0) and (12). However, we have since come to the conclusion that we cannot accurately determine the triplet p scattering length, due to the large uncertainties in the basis choice (see answer to (2) and a new paragraph in the Discussion section) and the relative insensitivity to this scattering channel. We have therefore opted to employ previously measured values instead.

(13) What is the exact procedure of extrapolating the scattering length to $k=0$?

- The scattering length for each k is calculated by numerically solving the Schrödinger equation for the model potential and analyzing the phaseshift between the resulting wavefunction and the wavefunction for a free particle. Meanwhile we have improved the numerical accuracy of this procedure and found that increasing the upper integration boundary gives a more accurate result for very small k (as the wavelength tends to infinity). Additionally, we became aware that there is an analytical solution for the zero-energy scattering length for the model potential we use (O'Malley et al. 1963 [3]). With the improved accuracy our calculation agrees with the analytical solution and we have altered the zero-energy value accordingly to $-14.6 a_0$ and added a citation for the analytical solution. Importantly, the improved extrapolated value does not alter any other part of the paper, as it is only relevant for very small values of k . We have also become aware of another paper (MacLennan et al. 2019 [4]), which measured the scattering length using D-state Rydberg molecules and got a similar result of $-14.7 a_0$. This value was added to the comparison with other measurements.

(14) I think the majority of the paragraph (Line 182 - Line 197) is actually discussion for future studies rather than the result. I suggest you move this part (or maybe the entire paragraph line 151- line 197) to Discussion section and merge it to the partially overlapping sentences (Line 268 - Line 281)

- We have merged this section into the Discussion section.

(15) Line 204: Could you provide numbers for the splitting and the resolution?

- The rotational splitting is about 60 kHz, while the FWHM of the molecular peaks is on average about 7 MHz. We have added this information to the paper.

(16) Line 215: How do you calculate the coefficient $C_i(R)$? Is this the same procedure as what you used to calculate the PEC?

- The coefficients are indeed the result of the diagonalization.

(17) Line 236- Line 238: Could you explain what you meant by this sentence? I assume TOF (undefined) stands for time-of-flight.

- When tunneling into the butterfly curve occurs, the perturber atom and Rydberg core are accelerated towards each other. At small internuclear distance they enter the regime of "classical" molecules, where the 5S electron is shared between the two cores. The exchange of the electron between the two cores can be thought of as a oscillating dipole which can cause transitions of the Rydberg electron. These transitions can also go to the continuum, in which case Rb_2^+ is formed. Because this molecular ion is heavier than the Rb^+ we get from photoionization, it takes longer to reach the time-resolved detector and can be counted separately. We have changed TOF to time-of-flight for clarity.

(18) If the lifetime is indeed affected by the tunneling, do you see an increase in the signal for ions with larger momentum? (Maybe you didn't collect this signal, but I am curious.) I imagine the rate of this tunneling (rate) is sensitive to v near the top of the potential well. Is it possible to use this to check the well depth?

- Yes, there is a definitive increase in ions with larger momentum for higher vibrational states. However, a quantitative analysis of this signal is difficult. The main reason for this is that after the most likely state-changing collision the atoms leave the ionization volume in about $2 \mu\text{s}$. For collisions into even lower states this happens even faster. To clearly separate the excitation pulse (which also creates ions) and ionization pulse they need to be at least $2 \mu\text{s}$ apart. This results in considerable uncertainties, but it is clear from the signal that tunneling plays a significant role in the decay of the higher vibrational states. We have altered the paragraph to make it clear that the stated sixty percent increase in rate for l -changing collisions and Rb_2^+ was obtained through a different experimental analysis than the lifetime measurement.

(19) Table 1: The experimentally measured dipole moment seems to be consistently bigger than the theoretical values. Do you have any explanation for this?

- Looking at Eq. 3 there are multiple possible explanations:

$$(1) \quad \langle d \rangle = \int |\Phi(R)|^2 \sum_{i,j} c_i^{(R)*} c_j^{(R)} \langle i | \hat{d} | j \rangle dR.$$

One possible explanation is that the semi-classical approximation of the calculation of k is wrong and the trilobite well actually forms at a larger internuclear distance. The electronic state composition also seems to be a likely candidate, as any inappropriate assumption or approximation about the interactions leads to different coefficients $c_i^{(R)}$. Further measurements at different principal quantum numbers combined with a more sophisticated theory (Greene et al. 2023 [5]) will hopefully lead to more insight. Another possibility is that the dipole matrix elements used for the calculation are systematically too large. This is equivalent to

the theoretical wavefunctions for the high- l states being incorrect, which seems relatively unlikely.

(20) Line 291 How did you estimate the quality factor? Which numbers did you use for this estimate?

- The quality factor is the oscillation frequency divided by the decay rate, or $Q = \frac{|E(\nu_l=1) - E(\nu_l=0)| \cdot \tau}{\hbar} = 546 \text{ MHz} \cdot 2\pi \cdot 10 \text{ } \mu\text{s} \approx 34000$.

Reviewer 2:

* The authors comment that bond states are calculated with a “shooting method.” This is not a term I am familiar with. Perhaps they could use a couple of sentences to expand upon it.

- We have added some additional information in the methods section regarding the calculation of the vibrational states. We have also added an additional citation regarding the shooting method.

*The authors comment that they find “electric dipole moments of up to 1735 Debye.” Given the sizable errors listed in Fig. 4, surely it would be better to state “up to ~ 1700 Debye.”

- We have changed the manuscript accordingly.

* The authors discuss resolving the discrepancy between previously-reported scattering lengths and their measurements by using different values of n . How easy is this to do? Would their existing lasers allow this? Such measurements would certainly strengthen the paper and further justify the use of long-range dimers for detailed study of electron-atom scattering.

- Regarding the discrepancy to other measurements we would like to add the following comment. The scattering length for each k is calculated by numerically solving the Schrödinger equation for the model potential and analysing the phaseshift between the resulting wavefunction and the wavefunction for a free particle. We have since improved the numerical accuracy of this procedure and found that increasing the upper integration boundary gives a more accurate result for very small k (as the wavelength tends to infinity). Additionally, we became aware that there is an analytical solution for the zero-energy scattering length for the model potential we use (O’Malley et al. 1963 [3]). With the improved accuracy our numerical calculation agrees with the analytical solution and we have altered the zero-energy value accordingly to $-14.6 a_0$ and added a citation for the analytical solution. Importantly, the improved extrapolated value does not alter any other part of the paper, as it is only relevant for very small values of k . We have also become aware of another paper (MacLennan et al. 2019 [4]), which measured the scattering length using D-state Rydberg molecules and got a similar result of $-14.7 a_0$. This value was added to the comparison with other measurements.

The laser that couples the 5D state to Rydberg states works for n values between 18 and 27. However, with the current theoretical model the error associated with the choice of basis is too large, such that no further insight could be gained from measurements in this n -range. A more sophisticated theoretical model, such as the one presented in Greene et al. 2023 [5] might provide smaller errors for future in-depth studies of the electron-atom scattering processes

REFERENCES

- [1] Matthew T. Eiles and Chris H. Greene. “Hamiltonian for the inclusion of spin effects in long-range Rydberg molecules”. In: *Phys. Rev. A* 95 (4 Apr. 2017), p. 042515. DOI: 10.1103/PhysRevA.95.042515. URL: <https://link.aps.org/doi/10.1103/PhysRevA.95.042515>.
- [2] F. Engel et al. “Precision Spectroscopy of Negative-Ion Resonances in Ultralong-Range Rydberg Molecules”. In: *Phys. Rev. Lett.* 123 (7 Aug. 2019), p. 073003. DOI: 10.1103/PhysRevLett.123.073003. URL: <https://link.aps.org/doi/10.1103/PhysRevLett.123.073003>.
- [3] Thomas F. O’Malley. “Extrapolation of Electron-Rare Gas Atom Cross Sections to Zero Energy”. In: *Phys. Rev.* 130 (3 May 1963), pp. 1020–1029. DOI: 10.1103/PhysRev.130.1020. URL: <https://link.aps.org/doi/10.1103/PhysRev.130.1020>.
- [4] Jamie L. MacLennan, Yun-Jhih Chen, and Georg Raithel. “Deeply bound ($24D_J + 5S_{1/2}$) ^{87}Rb and ^{85}Rb molecules for eight spin couplings”. In: *Phys. Rev. A* 99 (3 Mar. 2019), p. 033407. DOI: 10.1103/PhysRevA.99.033407. URL: <https://link.aps.org/doi/10.1103/PhysRevA.99.033407>.
- [5] Chris H. Greene and Matthew T. Eiles. *Green’s function treatment of Rydberg molecules with spins*. 2023. arXiv: 2308.02692 [physics.atom-ph].

Reviewers' Comments:

Reviewer #1:

Remarks to the Author:

Referee reply:

Dear Authors,

Thank you for taking the time responding to my previous comments. I appreciate your detailed, thorough reply. I hope my comments were helpful. I have only two minor comments at this point. (a) Regarding my previous comment (9), "confirm" is still a little too strong in my opinion because the observed data does not necessarily restrict the potential height quantitatively. I suggest something like "the observed highest levels are consistent with the theoretically predicted potential depth". (b) Re: line 167, I think you observed 7 levels for v_m series instead of 6. Maybe it is easier to say "observed up to $v=6$ ". I think you can also refer to Table 1 in this section.

Again, thank you for your efforts to update the manuscript. I am more than happy to recommend this manuscript for publication.

Sincerely,

Toshihiko Shimasaki, University of California Santa Barbara

Reviewer #2:

Remarks to the Author:

This revised manuscript answers the questions I raised in my earlier review. The paper is well written, well organized, and timely. The experimental results are of high quality and supported by theoretical calculations. The paper clearly merits publication. It is rather specialized and might be of limited general interest but I would not oppose its publication in Nature Communications.

Dear Reviewers,

we would like to thank you again for your invested time as well as for the useful questions and remarks about our paper that you have provided.

Regarding the two points raised by Reviewer 1: We have changed the text to the suggested phrasing.

Additionally we have added some information regarding the error bars in Fig. 4. All changes are highlighted in blue.

Sincerely

Max Althön, Markus Exner, Richard Blättner and Herwig Ott